# Acidic Microenvironment Enhances Cisplatin Resistance in Bladder Cancer via Bcl-2 and XIAP

**DOI:** 10.3390/cimb47010043

**Published:** 2025-01-10

**Authors:** Kaede Hiruma, Vladimir Bilim, Akira Kazama, Yuko Shirono, Masaki Murata, Yoshihiko Tomita

**Affiliations:** 1Department of Urology, Division of Molecular Oncology, Graduate School of Medical and Dental Sciences, Niigata University, Niigata 951-8510, Japan; vbilim@zoho.com (V.B.);; 2Department of Urology, Kameda Daiichi Hospital, Niigata 950-0165, Japan; 3Glickman Urological and Kidney Institute, Cleveland Clinic, Cleveland, OH 44195, USA; 4Department of Urology, Niigata Cancer Center Hospital, Niigata 951-8133, Japan; 5Department of Urology, Niigata Prefectural Central Hospital, Niigata 943-0192, Japan

**Keywords:** potential hydrogen (pH), acidity, bladder cancer, cisplatin, Bcl-2, XIAP, chloroquine, navitoclax, autophagy, apoptosis

## Abstract

Cisplatin (CDDP) remains a key drug for patients with advanced bladder cancer (BC), despite the emergence of new therapeutic agents; thus, the identification of factors contributing to CDDP treatment resistance is crucial. As acidity of the tumor microenvironment has been reported to be associated with treatment resistance and poor prognosis across various cancer types, our objectives in this study were to investigate the effects of an acidic environment on BC cells and elucidate the mechanisms behind CDDP resistance. Our findings show that BC cells cultured under acidic conditions developed cisplatin resistance as acidity increased. Notably, CDDP administered to BC cells in a pH 6.0 environment required double the concentration, compared to those in a pH 7.5 environment, to achieve equivalent toxicity. Using chloroquine and navitoclax, we identified the involvement of the Bcl-2 and LC3B pathways in the acquisition of CDDP resistance under acidic conditions. A Western blot analysis revealed that the activations of Bcl-2 and XIAP expression appear to inhibit both apoptotic and autophagic cell death. Taken together, these results suggest that alleviating the acidity of the tumor microenvironment in clinical settings might enhance BC sensitivity to CDDP.

## 1. Introduction

Bladder cancer (BC) was the 10th most diagnosed cancer type in the world in 2020, with 573,278 patient diagnoses and 212,536 deaths. In Japan, 34,568 patients were diagnosed and 10,928 died, with 8.7% of patients having had metastasis at the time of their initial diagnosis [1,2]. Among patients treated with transurethral resection of the bladder tumor (TURBT) and those diagnosed with pathological Ta and T1, the proportions of patients who progressed to muscle-invasive BC were 10% and 35%, respectively [3,4]. Metastatic recurrence is reported in 22.5% of patients with invasive BC who underwent curative TURBT [5].

Combination therapy including cisplatin (CDDP) has been a cornerstone of treatment for urothelial carcinoma, including urothelial BC, for decades. The most common platinum-based combination chemotherapy, gemcitabine and CDDP, was introduced in 1999, achieving an objective response rate of 48.4% to 49.1%; however, more than 80% of patients subsequently progress, resulting in a median OS of 12.8 to 14.0 months and a 5-year survival rate of 13%, leading to an unfavorable prognosis [2,6,7,8]. CDDP exerts its effects by binding to purine residues and damaging the deoxyribonucleic acid (DNA) of cancer cells, thereby inhibiting cell division and inducing apoptotic cell death; additionally, it generates reactive oxygen species that contribute to cell death [9].

There was no established standard of care following primary therapy with platinum drugs until the advent of pembrolizumab, which is typically administered either alone or in combination with docetaxel, paclitaxel, or vinflunine [10,11,12]. The JAVELIN Bladder 100 trial introduced avelumab maintenance therapy in patients with no disease progression after platinum-based chemotherapy [13,14]. Furthermore, the recent introduction of the antibody–drug conjugate enfortumab vedotin has expanded the options for BC treatment [15,16,17]. Despite recent advancements in various therapies, platinum-based drugs remain a key treatment, as a significant percentage of patients with advanced BC will use them during their lifetime. The tumor microenvironment has emerged as a factor in drug resistance; in several cancer species, growth inhibition and drug resistance have been associated with acidic environments [18,19]. In the context of BC, hypoxia has been identified as a trigger for drug resistance; however, there is little research addressing the impact of an acidic environment on this phenomenon [20,21,22].

Our main objective in this study was to investigate tumor microenvironmental factors that impair CDDP sensitivity in bladder urothelial cancer, with a specific focus on potential hydrogen (pH). To the best of our knowledge, this article is the first report on an association between an acidic environment and CDDP resistance in BC.

## 2. Materials and Methods

### 2.1. Cell Lines, Cell Culture, and pH Adjustment

We used the BC cell lines HT1376 and T24, as well as human embryonic kidney 293 (HEK293) cells, purchased from the American Type Culture Collection (ATCC, Manassas, VA, USA). HT1376 cells were cultured in RPMI 1640 medium (Gibco; Thermo Fisher Scientific, Inc., Grand Island, NY, USA) containing 10% fetal bovine serum (FBS; Gibco; Thermo Fisher Scientific, Inc.), 1% MEM non-essential amino acids (Gibco; Thermo Fisher Scientific, Inc.), 1% MEM sodium pyruvate solution 100 mM (Gibco; Thermo Fisher Scientific, Inc.), and 90 µg/mL of kanamycin at 37 °C in an atmosphere of 5% CO_2_. The potential hydrogen (pH) of the complete medium was adjusted to 0.5 intervals from pH 6.0 to pH 7.5 with hydrochloric acid, and the pH was measured with a pH monitor LAQAtwin (HORIBA, Kyoto, Japan). The amount of HCl required to reduce the pH of 40 mL of culture medium is shown in Appendix A. CDDP was purchased from WAKO, Osaka, Japan.

### 2.2. Cell Viability Assay

Cell viability was evaluated using an MTS assay with a tetrazolium compound according to the manufacturer’s instructions. Cells were seeded in 96-well plates at 2 × 10^3^ cells/well, incubated in 100 μL of unadjusted medium for 3 h, and then treated with CDDP for 0, 24, 48, and 72 h in each pH environment, after which 10 µL of CellTiter 96^®^ AQueous One Solution Reagent (Promega Corporation, Madison, WI, USA) was added to each well and incubated at 37 °C for 2 h. The absorbance was quantified at 490 nm using an IMARK microplate reader (Bio-Rad Laboratories, Inc., Hercules, CA, USA). Chloroquine (phosphate) (14194) was purchased from the Cayman chemical company (Ann Arbor, MI, USA). Inhibitory concentration 50 (IC50) was calculated using GraphPad Prism software (GraphPad Software version 8.0, Inc., San Diego, CA, USA). Data were analyzed using one-way ANOVA with Dunnett’s test for multiple comparisons. Statistical analysis was calculated using GraphPad Prism software.

### 2.3. Cell Migration Assay

Cells were spread at full confluence in 6-well plates and scratched with a p1000 pipet tip after 3 h of culturing, then washed with PBS, and incubated for 24 h in pH 6.0 and pH 7.5 environments, either with or without CDDP administration. Scratch assay area measurements were calculated using the software ImageJ (version 1.54 h) (NIH, Bethesda, MD, USA).

### 2.4. Western Blotting

Subconfluent cell cultures were washed with cold PBS and lysed using Cell Lysis Buffer (Cell Signaling Technology, Inc., Danvers, MA, USA) containing Protease Inhibitor Cocktail (Sigma-Aldrich, St. Louis, MO, USA). The cell lysates were incubated on ice for 30 min, then the lysates were clarified via centrifugation at 15,000× *g* for 30 min at 4 °C, and the supernatants were collected. The protein concentration was determined using the Bradford method. Each 20 µg of protein was separated on 4–15% SDS-polyacrylamide gel and transferred to a PVDF membrane. After blocking with 10% skimmed milk in TBS, the membranes were incubated with primary antibodies, and then with horseradish peroxidase-labeled secondary antibodies for one hour at room temperature. Immunolabelling bands were visualized using the Clarity Max Western ECL Substrate (Bio-Rad Laboratories, Inc., Hercules, CA, USA). The following primary antibodies were purchased from Cell Signaling Technology, Inc.: SQSTM1/p62 (8025), LC3B (3868), Beclin-1 (3738), HIF1a (14179), Cleaved PARP (9541), β-Actin (3700). BCL2 monoclonal antibody was purchased from Abnova, Inc., Taipei, Taiwan (MAB11332). XIAP was purchased from Proteintech, Inc., Rosemont, IL, USA (MAB11332). Anti-mouse IgG, HRP-linked Whole Ab Sheep and Anti-mouse IgG, HRP-linked Whole Ab Sheep were purchased from Cytiva Inc., Tokyo, Japan (NA931/NA934). The dilution ratios of the primary antibodies were 1:500–1:2000.

### 2.5. Flow Cytometry Analysis

For the cell cycle assessment, the cells were harvested 24 h after CDDP administration. The cells were washed twice with cold PBS and fixed in ethanol at 4 °C for 30 min, after which 1000 µL of FxCycle™ PI/RNase Staining Solution (Invitrogen; Thermo Fisher Scientific, Inc., Grand Island, NY, USA) was added and the cells were incubated at room temperature in the dark for 30 min. Flow cytometry was performed to assess the cell cycle using a BD FACSCelesta™ Flow Cytometer (BD Biosciences, Franklin Lakes, NJ, USA).

Data analysis was performed using the FCS Express™ (version 7.22.0031) (De Novo Software, Pasadena, CA, USA).

### 2.6. Confirmation of Apoptotic Cells

The bladder cancer cells treated for 72 h were detached from the flask using 0.05% Trypsin-EDTA (Thermo Fisher Scientific Inc., Grand Island, NY, USA), followed by thorough washing with PBS. They were then applied onto coated glass slides. After allowing the cells to air-dry sufficiently, they were fixed with 4% Paraformaldehyde Phosphate Buffer Solution (Fujifilm Wako Inc., Osaka, Japan) and stained with hematoxylin.

### 2.7. Statistical Analysis

All statistical analyses were conducted using GraphPad Prism software (version 10.3) (GraphPad Software, Inc., San Diego, CA, USA). Continuous variables are presented as mean ± standard deviation (SD). When there was one factor, a one-way ANOVA was used, and when there were two factors, a two-way ANOVA was used for multiple comparisons with Tukey’s test. Statistical significance was determined when the two-sided *p*-value was less than 0.05.

## 3. Results

### 3.1. BC Cells Become More Adaptive in Acidic Environments Compared to Normal Cells, Showing Strong Cisplatin Resistance Under Acidic Conditions

First, HEK293, T24, and HT1376 cells were cultivated for 72 h in media adjusted to pH 6.5, pH 7.0, and pH 7.5, respectively, in order to investigate the differences in acidic tolerance levels between normal cells and BC cells. Normal cells were severely inhibited in proliferation when exposed to an acidic environment, whereas BC cells showed resistance; in particular, HT1376 cells were more robust against acid toxicity than T24 cells, so in our subsequent experiments, we used HT1376 cells and added a more acidic medium with pH 6.0 (Figure 1).

HT1376 cells were conditioned in culture environments ranging from pH 6.0 to pH 7.5 and incubated for 24 h, showing a tendency to proliferate in normal environments compared to acidic environments; however, the administration of CDDP did not influence the growth of HT1376 cells cultured in the acidic environment, whereas it caused cell growth inhibition in the neutral environment (Figure 2a). Subsequently, HT1376 cells were cultured in pH 6.0 and pH 7.5 culture media, and various CDDP concentrations were added to the cells. The CDDP IC50 for HT1376 cells cultured in the acidic environment was 4.268 µM, which was twice that for the cells cultured in the neutral pH environment; based on these results, a basic dose of 2 µM of CDDP was used in subsequent experiments (Figure 2b). The aforementioned results show that an acidic environment has some positive effects on CDDP resistance in BC cells.

### 3.2. Acidic Environment Inhibits CDDP Treatment-Induced Mobility of BC Cells

In the migration assay, cell mobility was not significantly different under acidic conditions compared to normal conditions, whereas CDDP treatment impaired mobility capacity only under neutral pH conditions; thus, an acidic environment confers an advantage to BC cells not only in terms of survival, but also in terms of mobility (Figure 2c,d).

### 3.3. Bcl-2 Expression Is Activated and Autophagy Is Suppressed in BC Cells Under Acidic Conditions

To check the influences of autophagy and apoptosis on the survival of BC cells under low pH conditions, we performed a survival assay using the autophagy inhibitor chloroquine and the bcl-2 inhibitor navitoclax, aiming to determine which pathways influence CDDP resistance under acidic conditions. When chloroquine was administered under neutral pH conditions, HT1376 growth was inhibited in a concentration-dependent manner, with or without CDDP administration; this finding was not observed in an acidic environment. BC cells cultured in an acidic environment demonstrated decreased survival when treated with navitoclax, but this effect was not observed under neutral pH conditions, suggesting that the inhibition of autophagy in urothelial carcinoma cells in acidic environments may be linked to decreased survival (Figure 3).

The cell cycle analysis using flow cytometry showed no changes due to acidity under non-drug treatment conditions. CDDP treatment caused bladder cancer cells to be arrested in the S and G2/M phases, with a particularly high incidence of S phase arrest under neutral pH conditions (Figure 4).

Hematoxylin staining revealed morphological abnormalities in cells treated with CDDP. In particular, apoptotic bodies were observed in the CDDP-treated group under neutral pH conditions. In the acidic environment, although anticancer drug treatment caused cellular damage, the morphology of the cytoplasm and nucleus was relatively preserved. No significant morphological abnormalities due to changes in pH were observed in cells without CDDP treatment (Figure 5).

### 3.4. High Bcl-2 and XIAP Expression Levels in Acidic Environment Suppress Apoptosis and Autophagy-Induced Death in BC Cells

Subsequently, we studied the expression levels of apoptosis-related proteins using Western blotting. Bcl-2 expression was stronger in an acidic environment and was attenuated by the addition of CDDP. Cleaved PARP is the final step in the apoptosis cascade, and showed higher expression in the acidic environment without CDDP; on the other hand, when CDDP was administered, expression was activated in the acidic environment as well but showed stronger expression at a neutral pH level. In neutral pH conditions, the expression of caspase3 was reduced following CDDP treatment. In contrast, in acidic pH conditions, the expression of caspase3 was low prior to CDDP treatment, and CDDP caused a slight increase in its expression. Cleaved caspase3 levels mildly increased under neutral pH conditions following CDDP treatment; however, no increase was observed under acidic conditions. XIAP was expressed more strongly under acidic conditions and was suppressed by CDDP treatment in both environments. Since chloroquine was effective only in the neutral pH medium, we investigated the expression of LC3B, a protein involved in the formation of the autophagosome membrane, which was elevated in the neutral pH environment and increased slightly further with CDDP administration. The expression of P62, a protein that directs unwanted intracellular material toward autophagosomes, was elevated in acidic environments and slightly decreased with CDDP treatment. Beclin1 expression was not influenced by the acidity of the culture. HIF1α expression did not change significantly with pH differences, but a mild increase in expression was observed when CDDP was administered in an acidic environment (Figure 6).

## 4. Discussion

In normal tissues, extracellular pH is maintained at approximately 7.4; however, the pH around tumor cells decreases to between 6.7 and 7.2 [23,24]. Cells located at a distance from blood vessels may experience even greater acidification, potentially dropping to pH 5.6 due to the accumulation of lactate and protons [23,25]; a well-known contributing factor to this is a phenomenon known as the Warburg effect, in which cancer cells hyperactivate their glycolytic system even in an oxygen-rich environment, leading to increased lactate production and subsequent acidification [26].

The urothelium can be influenced by urine pH because it reabsorbs urine into the bloodstream; this is likely true for urothelial carcinoma, which is constantly exposed to urine [27]. Acidic urine has been associated with poorer postoperative prognosis in both upper urinary tract cancer and BC [28,29]; additionally, a higher recurrence rate has been observed in non-muscle-invasive BC treated with intravesical mitomycin C in conjunction with lower urinary pH [21]. HIF1a-mediated autophagy has also been implicated in the development of gemcitabine resistance in BC [20]. Conversely, some studies suggest that acidic urine is associated with inhibited growth in BC cells [30]. In other cancer types, such as breast cancer and squamous cell carcinoma, growth inhibition has been reported in acidic environments, although drug resistance has also been reported to be enhanced [18,19].

Based on Figure 1 and Figure 2a, we conclude that bladder cancer cells exhibit greater resistance to acidic environments than normal cells, although they are not entirely unaffected. Furthermore, the use of a more strongly acidic environment, as shown in Figure 1, resulted in a more pronounced toxic effect, highlighting the enhanced toxicity associated with acidic conditions. In other words, while acidic environments exhibit toxicity towards bladder cancer cells, they simultaneously induce drug resistance. Notably, we also demonstrated, for the first time, that CDDP resistance is increased under acidic pH conditions. Furthermore, scratch assays showed that CDDP administration did not impair HT1376 cell mobility in an acidic environment. One hypothesis for this observed drug resistance is that the weakly basic nature of CDDP prevents it from effectively reaching its intracellular targets in an acidic environment [31]; however, given that oxidative stress is a primary mechanism of action for CDDP, further investigation into alternative resistance mechanisms is warranted [9].

In order to explore the mechanisms underlying drug resistance in BC under acidic conditions, we utilized the autophagy inhibitor chloroquine and the Bcl-2 inhibitor navitoclax. Navitoclax, an orally available Bcl-2 inhibitor first reported by Chang J et al., is known to induce apoptosis [32,33], while chloroquine, commonly used as an antimalarial agent, is known to inhibit autophagy by blocking the fusion of autophagosomes and lysosomes [34]. In our experiments, neither navitoclax nor chloroquine significantly impacted the proliferation of BC cells in acidic environments. In the cell cycle analysis using flow cytometry, CDDP treatment under neutral conditions resulted in a greater transition to the S phase compared to under acidic conditions, which we believe may be due to a higher incidence of DNA damage. Western blotting showed that Bcl-2 protein expression was indeed enhanced and that cleaved PARP, downstream from the cascade, was suppressed in an acidic environment. XIAP is the most important member of the inhibitor of apoptosis protein (IAP) family and is known for its inhibition of apoptosis through directly blocking caspases [35]. High XIAP expression has been linked to poor prognosis in a variety of cancer types, including bladder and breast cancers, and has been reported to cause resistance to chemotherapy and radiation therapy [36,37,38]. Our findings indicate that BC cells in acidic environments not only exhibit increased Bcl-2 expression, but also upregulate XIAP, suggesting the activation of multiple anti-apoptotic pathways. Bcl-2 is known to negatively regulate autophagy by binding to Beclin1 [39], which is phosphorylated by AMP-activated protein kinase to promote autophagosome maturation [40]. Light chain 3 (LC3) is a subunit of the neuronal microtubule-associated proteins (MAPs), MAP1A and MAP1B [41]. LC3B is a membrane protein, essential for autophagosome formation, and serves as a reliable biomarker for autophagic activity [42,43]. p62/SQSTM1 is responsible for recognizing ubiquitin chains and directing specific proteins and cell organelles to the sequestration membrane of the autophagosome [44]. Our results showed that BC cells cultured in an acidic medium exhibited decreased LC3B activity and elevated p62/SQSTM1 levels, reflecting a reduction in autophagosome formation and an accumulation of p62/SQSTM1. We anticipated that Beclin1, which mediates autophagosome formation, would decrease under acidic conditions; however, we observed no significant change in its expression, although a slight reduction was noted with CDDP treatment, which may be attributed to the activation of Beclin1 by HIF1α. The role of HIF1α in our experiments, in which oxygen concentrations remained unchanged, warrants further investigation. We summarize the effects of CDDP administered in an acidic environment on apoptosis and autophagy pathways in BC cells in the following diagram (Figure 7).

Our study suggests that the acidic microenvironment surrounding bladder cancer cells exacerbates prognosis. Based on this, normalizing the extracellular environment may improve therapeutic responsiveness. In fact, animal studies have reported that treatment of acidosis with oral citrate formulations, among others, elevated the pH around the tumor and led to improved therapeutic outcomes [45,46,47]. In clinical practice, it has been reported that alkalinization of urine with oral sodium bicarbonate enhances the therapeutic efficacy of intravesical mitomycin C therapy for superficial bladder cancer [48].

## Figures and Tables

**Figure 1 cimb-47-00043-f001:**
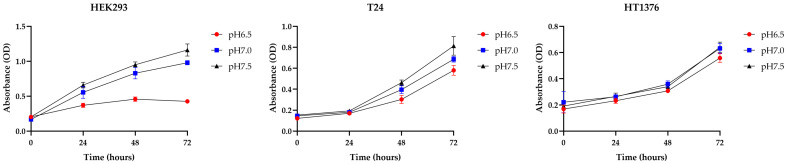
HEK293, T24, and HT1376 cell lines were subjected to survival assays to assess their responses to varying extracellular pH levels. Cells were plated in 96-well plates at a density of 2 × 10^4^ cells/mL and incubated for 72 h under different pH conditions. Low pH significantly inhibited the growth of normal HEK293 cells, while bladder cancer (BC) cells exhibited resistance to low pH. Notably, HT1376 cells were less sensitive to pH changes compared to T24 cells. Data are presented as mean ± SD from four independent experiments.

**Figure 2 cimb-47-00043-f002:**
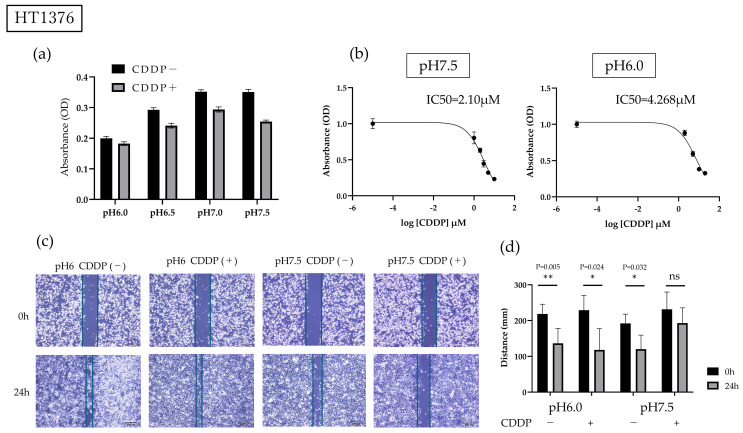
We studied the effects of pH and cisplatin (CDDP) on growth using the MTS assay and assessed migration inhibition in HT1376 cells. (**a**) Culturing HT1376 cells with 2 μM CDDP for 24 h in various pH environments revealed that higher medium acidity was associated with increased CDDP resistance. We analyzed the difference between groups with and without CDDP treatment using one-way ANOVA (*p* < 0.0001). (**b**) The CDDP IC50 was twice as high at low pH. (**c**) Bladder cancer cells were plated in 6-well flasks, scratched with a 1000 µL pipette tip, and incubated for 24 h. The green lines indicates the extent to which the cells have invaded. (**d**) The cell-free area was measured using ImageJ; bladder cancer cells grown in a neutral pH medium lost their migration ability when treated with CDDP, but those grown in an acidic pH medium did not. An asterisk (*) indicates that the *p*-value is less than 0.05, while two asterisks (**) indicate that the *p*-value is less than 0.01, ns indicates no significant difference.

**Figure 3 cimb-47-00043-f003:**
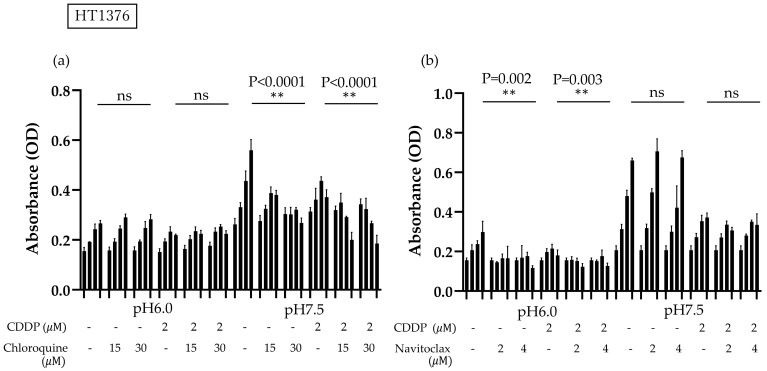
Effects of combinatory treatment with cisplatin (CDDP) and autophagy inhibitor chloroquine, or BCL-2 inhibitor navitoclax, at varying pH levels on cell viability. The time-dependent changes at 0 h, 24 h, 48 h, and 72 h of culturing are shown in each graph. CDDP was used at a concentration of 2 μM. Relative cell viability was assessed using the MTS assay after 72 h. Two asterisks (**) indicate that the *p*-value is less than 0.01, ns indicates no significant difference. (**a**) Chloroquine was administered at concentrations of 15 μM and 30 μM. Under neutral pH conditions, chloroquine inhibited HT1376 growth in a concentration-dependent manner, both with and without CDDP; this effect was not observed in an acidic environment. (**b**) Navitoclax was administered at concentrations of 2 μM and 4 μM. Bladder cancer (BC) cells cultured under acidic conditions showed reduced survival with navitoclax treatment, while no significant effect was noted under neutral pH conditions.

**Figure 4 cimb-47-00043-f004:**
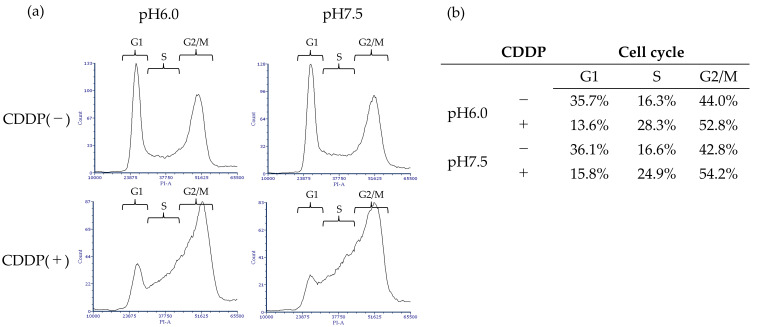
Cell cycle analysis using PI staining and flow cytometry. CDDP treatment led to arrest of the S and G2/M phases of the cell cycle in bladder cancer cells. The acidic environment had no significant effects on the cell cycle distribution. (**a**) This figure represents the ranges for the G1, S, and G2/M phases. (**b**) The proportions of each phase are quantified.

**Figure 5 cimb-47-00043-f005:**
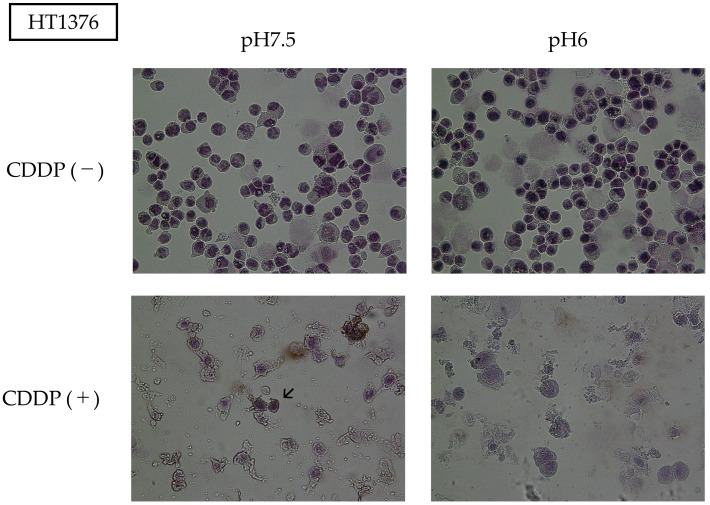
Optical microscope images (40×). In the absence of CDDP, the cells grew normally regardless of the pH. Treatment with CDDP caused cellular degeneration, and particularly at neutral pH, the appearance of apoptotic bodies was observed (see the arrows).

**Figure 6 cimb-47-00043-f006:**
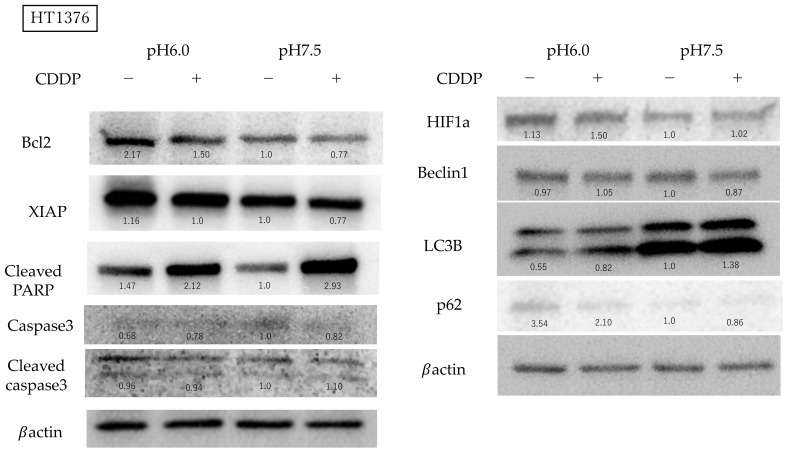
Western blot analysis comparing Bcl2, XIAP, cleaved PARP, Caspase3, Cleaved Caspase3, HIF1α, Beclin1, LC3B and p62 expression levels with β-actin as a loading control. We set protein expression under neutral conditions (i.e., in the pH 7.5 environment without cisplatin (CDDP) treatment) as 1.0 and quantified protein expression under other conditions. The original blots and ratios of protein expression to the βactin band are shown in Appendix A. BCL-2 expression was increased in an acidic environment and was reduced following CDDP treatment. CDDP decreased caspase3 expression and increased cleaved caspase3 expression in neutral pH. Cleaved PARP, a key marker of apoptosis, exhibited higher expression under acidic conditions without CDDP and increased expression under neutral pH conditions with CDDP. Additionally, XIAP expression was elevated under acidic conditions, but was diminished with CDDP treatment. LC3B levels were elevated under neutral pH conditions and increased further with CDDP treatment. P62 levels were higher in acidic environments and decreased slightly with CDDP administration. Beclin1 expression remained unaffected by pH levels. Additionally, the acidic environment promoted HIF1α expression.

**Figure 7 cimb-47-00043-f007:**
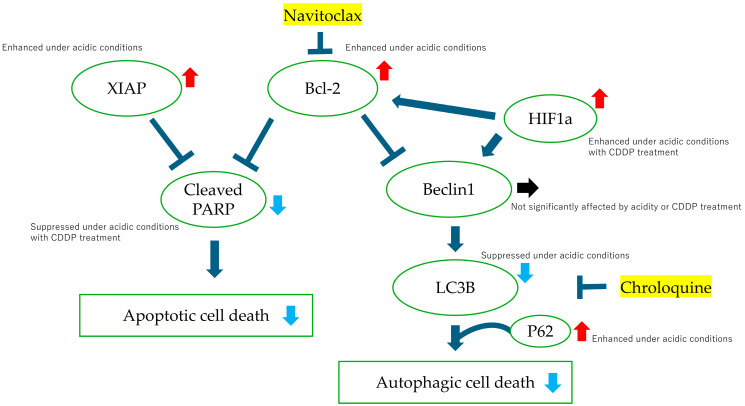
Graphical schema illustrating intracellular signaling pathways and the interactions of various proteins under acidic pH conditions with and without cisplatin (CDDP) treatment. Changes that we believe influence cisplatin resistance under acidic conditions are indicated with arrows. Red arrows represent enhancement, blue arrows indicate suppression, and black arrows denote no significant change. The schema highlights key proteins involved in apoptosis, autophagy, and hypoxia response, depicting how acidic environments and CDDP influence their expression levels and functional relationships.

## Data Availability

All data generated or analyzed during this study are included in this published article and its Appendix A.

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
