# Peer review of "Acidic Microenvironment Enhances Cisplatin Resistance in Bladder Cancer via Bcl-2 and XIAP"

_cimb, 2025, doi:10.3390/cimb47010043_

Round 1

Reviewer 1 Report

Comments and Suggestions for Authors

This manuscript investigated the mechanisms of acidic conditions promote Cisplatin resistance in bladder cancer via inhibiting apoptotic and autophagic cell death pathways. Overall, this is an interesting study and well-written manuscript. However, several concerns should be addressed before final publication.

Some of the statements in the manuscript are not clear. I suggest change the title of 3.2 to Acidic environment inhibits CDDP treatment-induced mobility of BC cells. And I suggest change the title of 3.4 to High Bcl-2 and XIAP expression levels in acidic environment suppress apoptosis and autophagy-induced death in BC cells.

In Fig2b, the inhibition of bladder cancer growth upon treatment of different dose of CDDP can be better presented using sigmoid curve after transform the CDDP concentration into log.

In Figure 3, the HT1376 cell growth is much quicker in the two untreated group under Ph7.5 compared with PH6.0. However, Figure 1 showed that acid environment did not inhibit the growth of HT1376 cells. How do the authors explain this apparent conflict observation?

Author Response

We sincerely thank you for your thoughtful message.

Comments 1: Some of the statements in the manuscript are not clear. I suggest change the title of 3.2 to Acidic environment inhibits CDDP treatment-induced mobility of BC cells. And I suggest change the title of 3.4 to High Bcl-2 and XIAP expression levels in acidic environment suppress apoptosis and autophagy-induced death in BC cells.

Response 1: We apologize for the confusion caused by the title and for any difficulty it may have caused in understanding the paper. We appreciate your suggestion for a clearer title and have made the change accordingly. We believe this revision has resulted in a more accessible manuscript.

Comments 2: In Fig2b, the inhibition of bladder cancer growth upon treatment of different dose of CDDP can be better presented using sigmoid curve after transform the CDDP concentration into log.

Response 2: We also value your input on how to improve Figure 2b. Following your recommendation, we have transformed the CDDP concentration into a log scale and plotted the sigmoid curve accordingly. Should there be any remaining unclear aspects or points requiring further revision, we would be most grateful for your valuable feedback.

Comments 3: In Figure 3, the HT1376 cell growth is much quicker in the two untreated group under Ph7.5 compared with PH6.0. However, Figure 1 showed that acid environment did not inhibit the growth of HT1376 cells. How do the authors explain this apparent conflict observation?

Response 3: We fully understand and agree with your comment regarding the inhibition of HT1376 cell growth in the experiment shown in Figure 3, as compared to Figure 1. We found no significant difference in cell growth between Figure 1 and Figure 3 under neutral pH conditions. Furthermore, at day 0, there was almost no difference in absorbance between the conditions under acidic environments. Based on these observations, we concluded that the difference in acidity between pH 6.5 and pH 6.0 affected cell growth after 72 hours. Specifically, while bladder cancer cells demonstrate some tolerance to acidic environments compared to normal cells, their growth is still inhibited as the acidity increases. We also observed the phenomenon of rapid growth inhibition with a pH difference of 0.5 between pH 7.0 and pH 6.5 in Figure 1.

Thank you again for your constructive feedback and continued support.

Sincerely,

Kaede Hiruma, M.D.

Reviewer 2 Report

Comments and Suggestions for Authors

Hiruma et al suggest that acidity of the tumor microenvironment in bladder cancer may be involved in the sensitivity to cisplatin. My comments are as follows;

In Figure 2, authors used HT1376 cells alone. Does acidity also affect proliferation of HEK293 cells?

Authors described that ‘’When chloroquine was administered under neutral pH conditions, HT1376 growth was inhibited in a concentration-dependent manner, with or without CDDP administration; this finding was not observed in an acidic 188 environment.’’

If so, I do not understand why the inhibition of autophagy in urothelial carcinoma cells in  acidic environments may be linked to decreased survival since chloroquine did not inhibit cell growth in acidic environments.

Authors should show data with statistical difference in Figure 4.

In Figure 5, authors should show data of immunohistochemical studies using some antibodies associated with apoptosis and autophagy such as cleaved caspase, annexin V and so on.

Author Response

We hope that, thanks to your feedback, our manuscript has become clearer.

Comments 1: In Figure 2, authors used HT1376 cells alone. Does acidity also affect proliferation of HEK293 cells?

Response 1: Thank you for your insightful comment regarding whether the experimental results in Figure 2 are also observed in normal cells. HEK293 cells exhibited marked growth inhibition under acidic conditions, as clearly demonstrated in the MTS assay, and were therefore not extensively used in subsequent experiments. However, in response to your valuable feedback, we are preparing to investigate the impact of acidic environments on the proliferation of normal cells. Despite our best efforts, we were unable to complete these additional experiments and obtain the necessary results within the revision period. We apologize for this delay and appreciate your understanding.

Comments 2: Authors described that ‘’When chloroquine was administered under neutral pH conditions, HT1376 growth was inhibited in a concentration-dependent manner, with or without CDDP administration; this finding was not observed in an acidic 188 environment.’’ If so, I do not understand why the inhibition of autophagy in urothelial carcinoma cells in  acidic environments may be linked to decreased survival since chloroquine did not inhibit cell growth in acidic environments.

Response 2: We are embarrassed by the elementary mistake I made with the terminology, and we sincerely appreciate your correction. We should have been more thorough in my review. Chloroquine inhibits the fusion of autophagosomes with lysosomes by alkalinizing the lysosomal environment. However, we believe that the lack of growth inhibition observed upon chloroquine treatment under acidic conditions is due to the inhibition of autophagosome formation.

Comments 3: Authors should show data with statistical difference in Figure 4.

Response 3: Thank you for your comment regarding the statistical differences in Figure 4. We are in the process of preparing the necessary data, but unfortunately, we were unable to complete it in time for the submission deadline. We sincerely apologize for this delay and hope for your understanding.

Comments 4: In Figure 5, authors should show data of immunohistochemical studies using some antibodies associated with apoptosis and autophagy such as cleaved caspase, annexin V and so on.

Response 4: Thank you very much for your valuable suggestion to include immunohistochemical experiments in Figure 5. In addition to the morphological changes, we conducted Western blotting using an antibody against cleaved caspase-3. We sincerely appreciate your meaningful advice and will certainly apply it to our future research.

Thank you again for your constructive feedback and continued support.

Sincerely,

Kaede Hiruma, M.D.

Reviewer 3 Report

Comments and Suggestions for Authors

 Dear authors,  

in the results section at lane 153 you indicated that HT1376 cells showed a tendency to proliferate in normal environments compared to acidic environments however in Figure 1  no difference in proliferation was observed at different pH.

 figure 2 reports CDDP IC50 could the author report the data as a percentage of the control

could the author perform a motility experiment in the presence of proliferation inhibitors?

Author Response

First and foremost, we would like to express our sincere gratitude for your valuable advice.

Comments 1: in the results section at lane 153 you indicated that HT1376 cells showed a tendency to proliferate in normal environments compared to acidic environments however in Figure 1  no difference in proliferation was observed at different pH.

Response 1: Bladder cancer cells exhibited a strong resistance to acidic environments, although they were not entirely unaffected. This became more evident when a more strongly acidic condition, pH 6.0, was introduced in the experiments. In Figure 1, HEK293 cells also seem to exhibit a stronger growth inhibition in response to environmental changes from pH 7.0 to pH 6.5, rather than from pH 7.5 to pH 7.0. Upon reviewing the graph in Figure 2(a), it appears that a similar trend may be occurring in bladder cancer cells as well. We apologize for any confusion caused by the unclear phrasing in our initial manuscript. We have added further clarification in the discussion section, and we believe that the revised version is now more comprehensible. We sincerely appreciate your constructive feedback.

Comments 2: figure 2 reports CDDP IC50 could the author report the data as a percentage of the control

Response 2: We apologize if my understanding is unclear. We observed that approximately twice the amount of anticancer drug was required to inhibit cell survival in acidic environments compared to normal environments. However, if there is a more appropriate way to express this, we would appreciate your guidance, and we will promptly make the necessary revisions. Additionally, based on feedback from another reviewer, we have revised the drug concentrations to a log scale for clarity.

Comments 3: could the author perform a motility experiment in the presence of proliferation inhibitors?

Response 3: We greatly appreciate your invaluable advice. In order to enhance the quality of our study, we are currently in the process of preparing results from scratch assays using chloroquine and navitoclax. However, we regret that we were unable to complete these experiments within the allotted time frame for submission, and we sincerely apologize for this delay.

Thank you again for your constructive feedback and continued support.

Sincerely,

Kaede Hiruma, M.D.

Round 2

Reviewer 2 Report

Comments and Suggestions for Authors

Authors modified the manuscript according to the comments, and I have no more comments.